# Can nighttime lights serve as a proxy for economic inequality at the local administrative unit scale? Evidence from Spain

Xaquín S. Pérez-Sindín[1,2]*, Piotr Wójcik[1], Tzu-Hsin Karen Chen[3,4], Alexander V. Prishchepov[2,5]

**1** Faculty of Economic Sciences, University of Warsaw, Warsaw, Poland, **2** Department of Geosciences and Natural Resource Management (IGN), University of Copenhagen, Øster Voldgade, København K, Denmark, **3** Department of Urban Design and Planning, University of Washington, University Wy NE, Seattle, Washington, United States of America, **4** Department of Environmental and Occupational Health Sciences, University of Washington, Nebraska, Seattle, Washington, United States of America, **5** Center for International Development and Environmental Research (ZEU), Justus Liebig University, Senckenbergstraße, Giessen, Germany

☕ These authors contributed equally to this work.
* xperez@wne.uw.edu.pl

## Abstract

Economic inequality remains a pressing issue, with localized disparities becoming increasingly visible. Monitoring these subnational dynamics is often constrained by the availability of timely, reliable data. This study evaluates the potential of nighttime light (NTL) data to proxy economic inequality at the scale of local administrative units (municipalities), using Spain as a case study. Spain is among the most unequal EU countries by the Gini index and has published municipal income distribution records since 2015. Its geographic and demographic diversity provides a robust setting for testing NTL-based measures with broader applicability. We combine two remote-sensing products (VIIRS and Harmonized NTL) with WorldPop gridded population data to compute lights-per-capita for each pixel. Within each municipality, Gini coefficients are calculated using pixels as the unit of analysis, enabling fine-grained spatial estimation of inequality. We derive Gini values for over 8,000 municipalities for 2015–2020 and compare them with official income-based Gini coefficients from Spain's tax administration. Relationships between light- and income-derived measures are assessed using cross-sectional regressions and multilevel panel models. Results show statistically significant associations between NTL- and income-derived Gini coefficients, particularly in smaller settlements and rural areas. Panel models outperform cross-sectional models, indicating that NTL data capture temporal changes in inequality even when contemporaneous correlations are modest. Some regions exhibit elevated NTL-derived inequality despite low official estimates—often where unemployment or informal economic activity is prevalent—suggesting that light-based metrics detect disparities not fully reflected in formal income statistics.

**Data availability statement:** Dataset, images and Rstudio code for all the analysis is available at https://github.com/Xaquin05/NTLInequality.git.

**Funding:** Financial support received from the University of Warsaw as part of the IDUB program (Action I.3.9 "Human mobility and inequalities as seen through digital data sources") is gratefully acknowledged. Th European Commission's Marie Skłodowska-Curie Individual Fellowship Action supported this work [Grant Agreement number: 101067663 — MAPSOCEXTRACT]. The funders had no role in study design, data collection and analysis, decision to publish, or preparation of the manuscript.

**Competing interests:** The authors declare that they have no known competing financial interests or personal relationships that could have appeared to influence the work reported in this paper.

These findings demonstrate the promise of NTL-derived indicators as scalable, low-cost, and globally replicable tools for monitoring inequality in both data-rich and data-poor settings. As subnational disparities gain importance, this framework offers a practical avenue for fine-grained and inclusive socioeconomic monitoring. Future work should integrate additional spatial layers—such as urban form, infrastructure, and demographics—to refine accuracy and interpretation.

## Introduction

Following the early 2000s financial liberalization and the 2008 global financial crisis, the discourse surrounding economic inequality has gained significant momentum, largely driven by its implications for economic performance [1], and democratic stability [2,3]. While global data suggest a narrowing gap between the richest and poorest nations, with the income disparity between the wealthiest 10% and the poorest 50% decreasing from approximately 50 times to just under 40 times, internal disparities within countries have markedly increased [4]. This paradoxical trend underscores the complexity of inequality as a multifaceted phenomenon that transcends simple economic indicators to incorporate regional and local disparities.

The growing interest in understanding the nuances of inequality has underscored the importance of having granular data at more localized geographic levels. Conventional sources provide economic indicators at national scales [5–8] and even at regional levels through datasets like the Luxembourg Income Study [9]. Yet, data for Local Administrative Units (LAUs) or municipalities is often limited, hindering more localized and detailed analyses of inequality. This is regrettable, as recent studies indicate that the perception of economic conditions, including the degree of inequality, is influenced by everyday life, which occurs at the local level. Research indicates that inequality at the local level of residence provides a stronger explanation of individual financial well-being than broader state-level measures [10]. Moreover, individuals exposed to greater inequalities within their immediate social environments tend to exhibit lower tolerance for such disparities [11]. Therefore, advancing our understanding of the causes and implications of inequality at a local level requires the development of innovative methodologies.

Nighttime lights (NTL) recorded with satellite remote sensing emerges as a promising proxy for economic studies [12–14]. While individual households cannot be directly observed from satellite images, the spatial clustering of wealthier and poorer neighborhoods suggests that variation in NTL emitted per capita could reflect economic inequality. Previous studies have demonstrated a significant correlation between NTL-based estimates and traditional measures of inequality, such as Gini coefficients of income [15–22]. This relationship is observed not only across countries but also within smaller regions, such as U.S. states [16]. Research further validated the use of NTL-derived Gini coefficients at the local level, showing strong alignment with estimates from African Demographic and Health Surveys (DHS) data [18]. However, these findings are limited to a single year, a specific sample of African countries, and regions with available DHS data.

Using Spain as our case study, we assess the potential of nighttime lights (NTL) as a predictor of spatial and temporal variations in economic inequality at the local level. Spain exhibits one of the highest Gini coefficients of income among the European Union state members, making it an interesting case study. Moreover, Spain's comprehensive administrative records on income distribution since 2015 provide a robust benchmark for validating NTL-derived measures of inequality. Finally, one unique aspect of Spain is the presence of large, sparsely populated areas [23], reflected in the significant proportion of municipalities with fewer than 2,000 inhabitants. This setting provides valuable insights for applications in other countries, particularly those that, like some parts of Spain, are less urbanized compared to highly urbanized areas in much of Western Europe.

To calculate NTL-derived Gini coefficients, we first estimated lights per capita by combining the NTL intensity data with gridded population estimates from the WorldPop dataset [24,25]. WorldPop, available for the years 2000–2020, offers high-resolution population distributions at a spatial resolution of 100 meters, derived from census data, satellite imagery, and advanced modeling techniques. By aligning the NTL data with these population estimates, we calculated per capita light intensity for each pixel, forming the basis for deriving the NTL Gini coefficients. To estimate the lights per capita for each of the pixels, we utilized high-resolution satellite data from two sources: VIIRS NTL [14,26,27] and Harmonized NTL [28]. The VIIRS NTL dataset, with a spatial resolution of approximately 750 meters, provided annual observations of artificial light intensity from 2012 onward. The Harmonized NTL dataset, which integrates data from VIIRS and the DMSP-OLS satellite systems, offered a consistent framework for comparing light intensity over a longer period time (1992–2021), albeit at a coarser spatial resolution of 1 kilometer.

We addressed two specific questions using Spain as our study area. First, do Gini coefficients derived from NTL per capita (NTL-derived Gini) correlate with those calculated from income per capita data (Income-derived Gini) across municipalities? Second, can annual variations in NTL-derived Gini coefficients reliably predict annual changes in income-derived Gini coefficients across municipalities?

Using NTL-derived Gini as independent variable and income-derived Gini as dependent variable, we conducted two distinct regression models to answer our research questions: a cross-sectional bivariate lineal regression analysis for the year 2015, 2016, 2017, 2018, 2019 and 2020 and one panel-data regression model for the study of temporal inequality dynamics between 2015 and 2020. Both models were run separately for each of the NTL products (VIIRS and Harmonized NTL). To ensure the robustness of our results and minimize the influence of outliers, we performed separate analyses based on municipality size, grouped by population thresholds.

## Materials and methods

### Study area

Spain is situated in the southwestern part of Europe, covering a land area of 505,990 km$^2$. As of 2023, it is estimated to have a population of approximately 47 million people, making it the fourth-most populous country in the European Union, following Germany, France, and Italy. Spain's administrative divisions consist of autonomous communities, provinces, and municipalities, corresponding to the European Union's Nomenclature of Territorial Units for Statistics (NUTS) level 2, level 3 and Local Administrative Units (LAUs), respectively. There are 17 autonomous communities and 2 autonomous cities, further divided into 50 provinces and over 8,000 municipalities (Local Administrative Units, LAUs) with an average area of 7,325 hectares (approximately 4.83 kilometers of radius).

Spain's population grew from 30 million in 1960 to 47 million in 2023. However, this growth occurred unevenly. The country underwent a rapid and significant process of population concentration in more urban areas, with cities like Madrid and Barcelona doubling their population since 1990 [29,30] (Fig 1). In contrast, rural areas in Spain have witnessed one of the most pronounced declines within the European Union [31]. According to classifications by the World Bank, the percentage of Spain's rural population in 2022 stood at 19%, compared to 25% in 1990 and 43% in 1960. Out of a total of 8,000 municipalities in Spain, 85% have fewer than 2,000 inhabitants, of which a significant portion (N = 1,396) barely

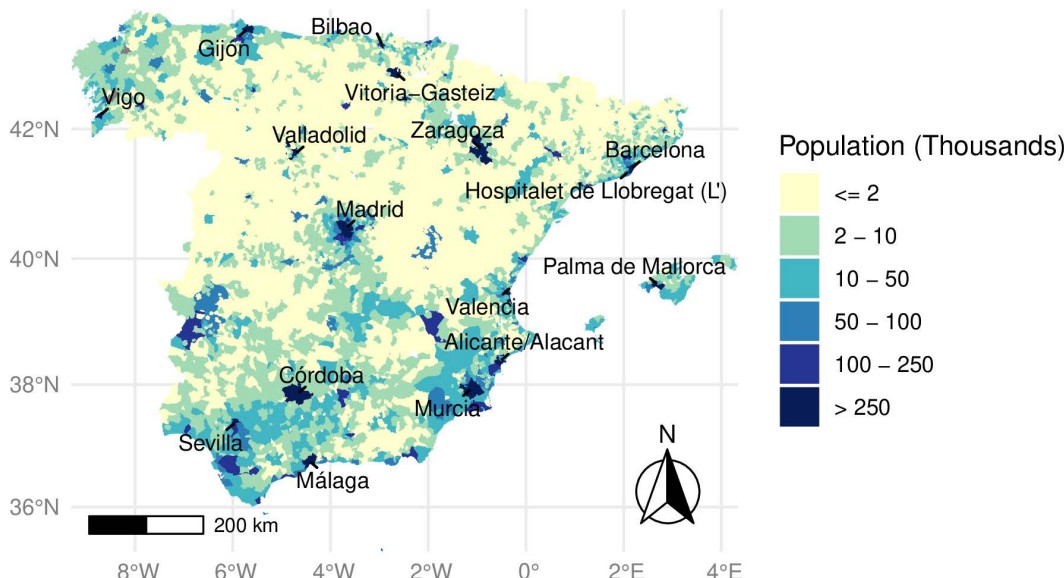

**Fig 1. Map of Spain by municipalities and population size and the name of cities with more than 250,000 inhabitants.** Source: Municipal Population Register, National Statistics Institute (INE) [32] Map created by the authors using publicly available data and Digital Cartography Files from (INE) [32]. This figure is published under the terms of the Creative Commons Attribution License (CC BY 4.0).

ranges between 0 and 100 residents (S1 Fig). This statistic underscores the extensive presence of small settlements across the country, highlighting the stark contrast between the densely populated and compact urban centers and the declining and sparsely inhabited rural areas.

Spain's Gini coefficient in 2020 stood at 32.1 [33], compared to the European Union average of 30. During the period studied here, particularly from 2015 to 2020, Spain's Gini coefficient saw a fluctuating yet insightful trajectory. It decreased from 34.6 in 2015 to 32.1 in 2020 (S2 Fig), suggesting a gradual narrowing of income disparities, potentially reflecting the impacts of economic recovery and policy measures aimed at reducing inequality after the 2008 financial crisis.

### Data sources

**Acquisition of benchmarking data on inequality.** We rely on data on Gini coefficient from the Atlas of Household Income Distribution (ADRH) [34] produced by the National Statistics Institute of Spain (INE) [32]. The Gini coefficient is a statistical measure used to quantify income inequality within a population, with values ranging from 0 to 1. A Gini value of 0 represents perfect equality, where all individuals receive the same income, while a value of 1 indicates total inequality, where one person holds all the income. ADRH offers data for all Spanish municipalities between 2015 and 2021. To calculate the Gini index, the INE relies on administrative records, specifically on income data provided by the Spanish Tax Agency and regional tax authorities (Haciendas Forales). This approach allows the calculation of a reliable indicator without relying on surveys, as the Gini coefficient is derived from the adjusted net income of households. A key factor in this process is the modified OECD scale, used to calculate income per consumption unit. This scale adjusts income based on household size and composition, assigning a weight of 1 to the first adult, 0.5 to additional members aged 14 or older, and 0.3 to those under the age of 14 [34]. In other terms, children are weighted less than adults when computing per-capita income.

ADRH relies on income data from tax returns provided by the Spanish Tax Agency and regional authorities. As a result, it does not capture undeclared income, such as that from informal work, potentially underestimating income levels in areas

or sectors where informal employment is common. Second, ADRH excludes certain pensions and income received from abroad by foreign residents, which affects areas with high populations of European retirees or other foreign residents. This exclusion can skew income distribution in these specific regions. Finally, despite ADRH adjusts for the highest and lowest 1.5% of incomes to minimize outliers, there remains a potential margin of error in areas with few tax filers or smaller populations. This limitation is particularly relevant in small municipalities or census tracts where data may not fully represent local income variation.

**Nighttime lights data.** The VIIRS NTL dataset is produced by the National Aeronautics and Space Administration (NASA) and the National Oceanic and Atmospheric Administration (NOAA) [14]. It is based on observations from the Suomi National Polar-Orbiting Partnership (Suomi NPP) and NOAA-20 satellites, specifically using the Visible Infrared Imaging Radiometer Suite (VIIRS) Day/Night Band (DNB). The VIIRS DNB captures detailed nighttime light intensity data, providing a global view of artificial lighting from space. This dataset is available annually and covers the years 2012 to the present. One of the significant advantages of the VIIRS product is its higher spatial resolution, which is typically 750 meters at the equator. This allows for finer granularity in detecting and analyzing variations in light intensity, particularly in urban areas where light intensity can change rapidly over small distances. However, the VIIRS dataset also has certain limitations. It can sometimes exhibit negative values, which are typically the result of sensor errors, calibration issues, or interference from atmospheric conditions, such as cloud cover. These negative values do not represent real-world phenomena and were therefore excluded from the analysis. They were implicitly removed during rasterization and aggregation steps, and any remaining invalid values were omitted from calculations. Additionally, the VIIRS dataset can experience some data anomalies in areas with frequent cloud cover, which might affect the accuracy of light intensity measurements.

In contrast, the Harmonized NTL dataset is an integrated product that combines data from multiple satellite sources, including the VIIRS DNB from the Suomi NPP and NOAA-20 satellites, and the DMSP-OLS (Defense Meteorological Satellite Program – Operational Linescan System) from the U.S. Department of Defense [28]. This dataset is produced by NOAA and aims to provide a consistent, high-quality global nighttime light product by harmonizing the data from these two different satellite systems. The Harmonized dataset spans a significant temporal range, incorporating DMSP data from 1992 to 2013 and VIIRS data from 2012 onwards. The ability to merge these two datasets is crucial for conducting long-term analyses of nighttime light patterns, as it allows for the measurement of light intensity and inequality over a period of 30 years. This long historical range is particularly valuable for tracking trends in urbanization, economic development, and inequality over an unprecedented length of time. The harmonization process ensures that the data from both satellite sources are comparable and consistent, correcting for any discrepancies caused by differences in sensor calibration, resolution, and other technical factors. While the Harmonized dataset provides a global overview and a more consistent product, it comes with some trade-offs in spatial resolution compared to VIIRS. Typically, the Harmonized dataset has a 1-kilometer resolution, which, while sufficient for identifying broad patterns, may obscure finer-scale details in urban areas.

**Population data.** We utilized the WorldPop population dataset [24,25] to estimate lights per capita for each pixel in the NTL composites before being able to estimate the Gini coefficients. The WorldPop dataset provides high-resolution, gridded population count estimates at a 100-meter resolution for multiple years, spanning from 2000 to 2020. This dataset offers globally consistent population estimates that integrate various data sources, including national census data, survey data, and satellite imagery. One of the advantages of the WorldPop dataset is its high spatial resolution (100 meters), which aligns well with the spatial resolution of the VIIRS and Harmonized NTL datasets. The detailed population estimates allow for accurate calculations of lights per capita at the pixel scale. However, there are some limitations to the WorldPop dataset as well. For instance, in areas with poor-quality or missing data (e.g., conflict zones or remote rural regions), the population estimates may have a higher degree of uncertainty. Moreover, while WorldPop strives to update its estimates annually, some regions may have gaps or inconsistencies depending on the availability and accuracy of input data.

## Data processing

All the datasets were processed to ensure they aligned with the administrative boundaries of the municipalities in the study area using both GIS software QGIS and R studio. To achieve this, the datasets were reprojected to a common coordinate reference system (CRS), specifically EPSG:4326 (World Geodetic System 1984), which allows for consistency across the various data sources for Spain.

To estimate lights per capita for each pixel of the VIIRS and Harmonized NTL datasets, the following process was followed:

1. Population Extraction: The WorldPop [25] population dataset was overlaid on a vectorized NTL raster data, extracting the population value for each corresponding NTL pixel to calculate the population sum per pixel.

2. Excluding Non-Populated Areas: Areas with no population, such as commercial or industrial zones, were excluded to avoid biases caused by light emissions in uninhabited areas.

3. Calculating Lights per Capita: Lights per capita was calculated by dividing the total light intensity for each pixel by its population, using a logarithmic transformation to normalize the light data.

4. Rasterization: The calculated lights per capita values were rasterized, creating a new NTL composite for spatial analysis.

5. Gini Coefficient Calculation: The Gini coefficient was calculated based on lights per capita for each pixel within a municipality, with values ranging from 0 (perfect equality) to 1 (perfect inequality)

## Methods

**Cross-sectional modeling.** To analyze the relationship between NTL Gini (dependent variable) and NTL Gini (predictor or independent variable), we fitted a model using ordinary least squares (OLS) regression. The following formula represents the structure of our model:

$$\text{Income} - \text{derived Gini}_{it} = \beta_0 + \beta_1 \cdot \text{NTL} - \text{derived Gini}_{it} + \in_{it} \qquad (1)$$

- Where Income-derived Gini$_{it}$ represents the value of the dependent variable for municipality i in year t,

- NTL-derived Gini$_{it}$ is the predictor variable for municipality i in year t,

- $\beta_0$ is the intercept term, representing the value of Income-derived Gini when NTL Gini is zero for that year,

- $\beta_1$ is the coefficient of NTL-derived Gini, representing the effect of NTL-derived Gini on Income-derived Gini for each year,

- $\in_{it}$ is the residual for municipality i in year t, representing the unobserved factors affecting Income-derived Gini.

**Panel data modeling using a multilevel approach.** We used a multilevel mixed-effects model to assess the capacity of NTL to predict income inequality. The model was structured with Income-derived Gini coefficients as the dependent variable and NTL-derived Gini coefficients as the independent variable. To account for spatial and temporal dependencies, we included random intercepts and slopes for municipalities, specifying that each municipality could have a different baseline value (intercept) and different effect of NTL-derived Gini (slope). The model was fitted using maximum likelihood estimation.

The multilevel model with random intercepts and random slopes is structured as follows:

$$\text{Income} - \text{derived Gini}_{it} = \beta_0 + \beta_1 \cdot \text{NTL} - \text{derived Gini}_{it} + b_{0i} + b_{1i} \cdot \text{NTL} - \text{derived Gini}_{it} + \in_{it} \qquad (2)$$

where:

- Income-derived Gini$_{it}$ represents the administrative records-based Gini Coefficient in municipality i in year t.

- $\beta_0$ is the fixed intercept, capturing the average baseline level of Income-derived Gini.

- $\beta_1$ is the fixed slope of NTL-derived Gini, representing the overall effect of NTL Gini on Income Gini.

- b$_{0i}$ is the random intercept for each municipality, allowing for differences in baseline Income-derived Gini levels across municipalities.

- b$_{1i}$ is the random slope for each municipality, allowing the effect of NTL Gini on Income Gini to vary by municipality.

- $\in_{it}$ is the residual error term, representing year-level deviations.

This model formulation allows for municipality-specific variability in both the baseline Income-derived Gini coefficient levels and the impact of NTL Gini on NTL Gini. In particular, the inclusion of the random slope b$_{1i}$ enables us to account for differences in the relationship between NTL Gini and NTL Gini across municipalities, thereby capturing spatial heterogeneity in this effect.

The model also accounted for temporal correlation within each municipality by applying a Gaussian correlation structure based on year, assuming that measurements from the same municipality in consecutive years are more correlated than those further apart [35]. This correlation is modeled as a Gaussian function, where the correlation between years decreases with the time interval. Thus, for two years t and t′ in municipality i, we assume:

$$\text{cor}\left(\in_{it}, \in'_{it}\right) = \sigma 2 \exp(-\varphi|t - t'|) \tag{3}$$

where σ2 represents a constant serial variance, and ϕ is an association parameter. This temporal structure considers that measurements taken closer together in time are more strongly correlated than those further apart [35].

## Results

### Correlation between NTL and Income-derived Gini coefficients

The results indicate that both the VIIRS NTL and Harmonized NTL models show positive correlations with the dependent variable (Income-derived Gini) across all years (Table 1). The VIIRS NTL model consistently exhibits stronger explanatory power, with higher $R^2$ values and more significant estimates compared to the Harmonized NTL model. However, the $R^2$ values remain relatively low for both models, indicating limited explanatory strength.

Both NTL and Income-derived Gini coefficients show that the highest economic inequality levels are observed in central Spain (Fig 2), particularly in the highly urbanized Madrid area, as well as in other densely populated regions along the Mediterranean coast, in the south (e.g., Málaga), the northwest (metropolitan area of Vigo), and the north (Bilbao and Gijón). In contrast, the most sparsely populated areas show the lowest Gini coefficient. Municipalities shaded in grey on the Income-derived Gini map represent areas lacking available data, primarily due to their small population size. Nevertheless, the map representing our NTL-derived Gini coefficients indicates low Gini coefficients for these areas. This contrast highlights one advantage of NTL data—it provides continuous spatial coverage, including in rural or excluded areas, where traditional survey or administrative data is unavailable.

The greatest discrepancy between the two data sources appears in southern Spain, particularly along the Guadalquivir urban axis (Seville, Córdoba, and Jaén) (Observe the green strip located in the southwest region of the map in Fig. 2). This region is an urban area marked by low productivity and some of the highest unemployment rates in both Spain and Europe, with levels reaching up to 30%. According to the income data, the estimated Gini coefficient here generally shows low inequality levels, aligning with trends in other parts of southern Spain. However, Gini estimates derived from VIIRS

**Table 1. Bivariate linear regression results for Income-derived Gini and NTL Gini by year.**

| NTL VIIRS | | | |
|---|---|---|---|
| **Year** | **Intercept** | **NTL Estimate** | **NTL R²** |
| 2015 | 29.62*** | 2.48*** | 0.030 |
| 2016 | 29.79*** | 1.94*** | 0.017 |
| 2017 | 29.13*** | 1.39*** | 0.009 |
| 2018 | 28.46*** | 1.52*** | 0.012 |
| 2019 | 28.16*** | 1.08*** | 0.006 |
| 2020 | 28.44*** | 1.16*** | 0.007 |
| NTL Harmonized | | | |
| | **Intercept** | **NTL Estimate** | **NTL R²** |
| 2015 | 30.09*** | 1.92*** | 0.0057 |
| 2016 | 30.15*** | 1.61*** | 0.0040 |
| 2017 | 29.51*** | 0.58* | 0.0007 |
| 2018 | 28.84*** | 0.77** | 0.0013 |
| 2019 | 28.31*** | 1.16*** | 0.0034 |
| 2020 | 28.67*** | 0.66** | 0.0012 |

(***) for highly significant results (p-value<0.001), (**) for very significant results (p-value<0.005), (*) for statistically significant results (p-value<0.01), and a dot (.) for marginally significant results (p-value<0.1).

light data indicate the opposite: inequality levels comparable to those found in urban areas of central Spain. The inability of official data to capture these disparities may be related to the exclusion of the informal economy—an element likely more prominent in high-unemployment areas. In such cases, NTL data may act as a proxy for unregistered economic activity, reinforcing its utility as a complementary measure of inequality.

The Harmonized NTL model tends to outperform VIIRS in larger population categories like 100,000–250,000, where it demonstrates stronger predictive power and higher R² values (Table 2). Conversely, VIIRS NTL model excels in smaller population categories, such as 2,000–10,000 and 10,000–50,000, where it shows significant positive correlations, particularly in VIIRS Estimate values.

## Capacity of NTL to predict temporal inequality dynamics

The multilevel models for predicting changes in inequality based on night-time light (NTL) data, both from the VIIRS and Harmonized datasets, demonstrate robust overall performance (Table 3). Both models have high conditional R² values (0.75 for VIIRS and 0.73 for Harmonized), indicating that they explain a substantial portion of the variability in inequality when both fixed and random effects are considered. Additionally, the models display strong statistical significance for their key parameters (p-values<0.001 for both intercept and slope terms). This highlights that both VIIRS and Harmonized NTL data are effective predictors of changes in inequality, capturing meaningful spatial and temporal variations in the dataset.

Despite their strong overall performance, the two models exhibit some notable differences. The VIIRS model outperforms the Harmonized model slightly in terms of model fit, as indicated by its lower AIC and BIC, and higher log-likelihood (Table 3). The random effects structure also differs, with the VIIRS model showing higher variability in both the intercept (Std. Dev. 3.44 vs. 3.10) and slope (Std. Dev. 5.03 vs. 3.95). Additionally, the spatial correlation range is slightly larger for the VIIRS model (1.18 vs. 1.09), suggesting it captures spatial dependencies over a broader area. The fixed effect slope for NTL is stronger in the VIIRS model (4.33 vs. 2.41), indicating that changes in NTL measured by VIIRS are more strongly associated with changes in inequality. However, the Harmonized model has lower residual variation (Residual

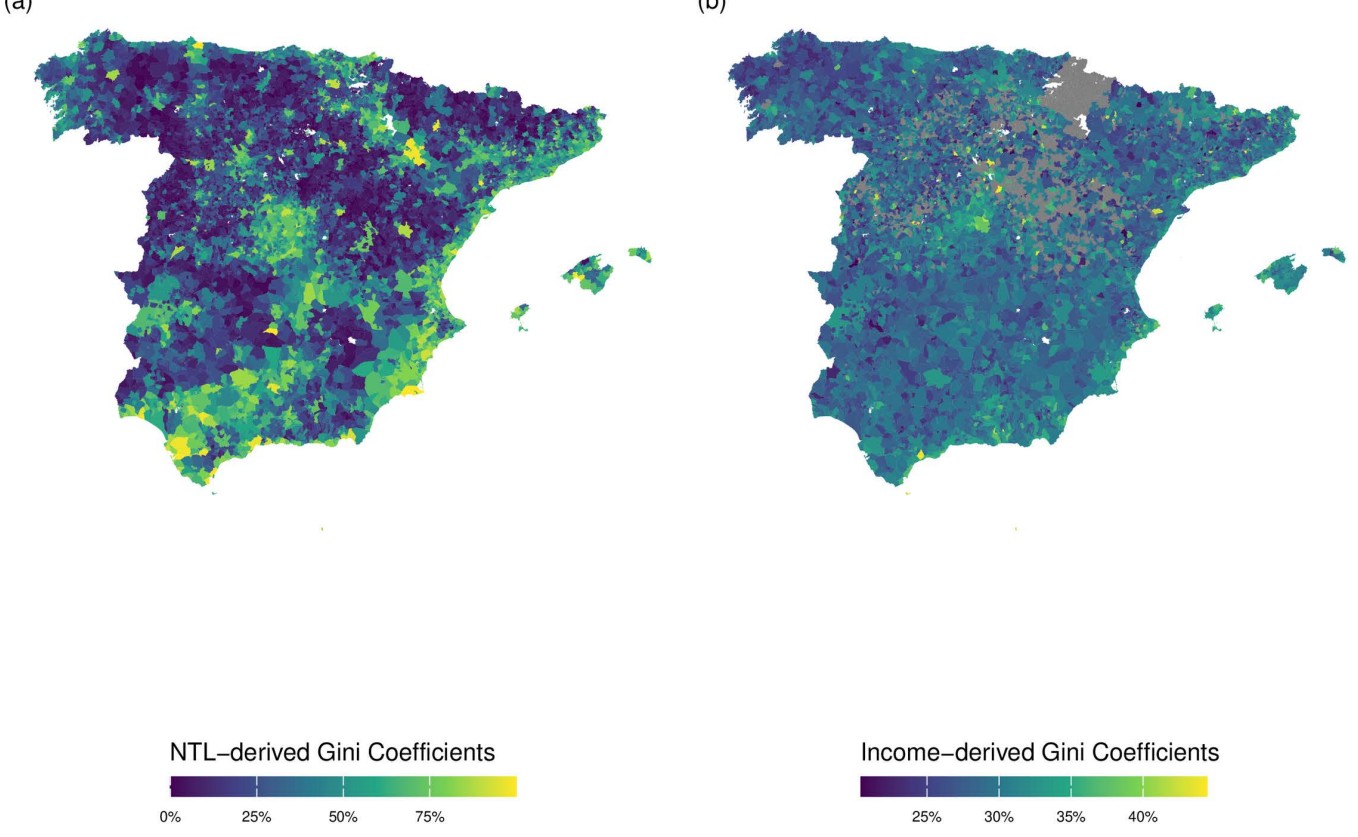

(a)

(b)

**NTL–derived Gini Coefficients**

0%  25%  50%  75%

**Income–derived Gini Coefficients**

25%  30%  35%  40%

**Fig. 2. Gini coefficients of Spanish municipalities (2020): (a) VIIRS NTL-derived coefficients and (b) Income-derived coefficients.** Source: VIIRS NTL-derived coefficients is based on data produced by the National Aeronautics and Space Administration (NASA) and the National Oceanic and Atmospheric Administration (NOAA) [14]. Income-derived coefficients are produced with data from Atlas of Household Income Distribution (ADRH) [34] produced by the National Statistics Institute of Spain (INE) [32]. *Municipalities shown in grey indicate missing data due to the fiscal autonomy of Spain's foral territory (Navarra), where income statistics are managed by regional tax authorities and are not included in the national tax authority's dataset. It can also be due to municipalities whose population falls below the national publication threshold.* Maps created by the authors using publicly available data and Digital Cartography Files from (INE) [32]. This figure is published under the terms of the Creative Commons Attribution License (CC BY 4.0).

Std. Dev. 1.83 vs. 1.86), reflecting potentially more stable predictions. In conclusion while both datasets perform well, the VIIRS model may be slightly better at capturing variability and spatial patterns in inequality changes.

Fig 3 provides evidence that NTL-derived Gini correlates with NTL Gini time-series at a higher rate in the more urban areas. The positive mean random slopes in the municipalities with a population below ten and five thousand suggest that NTL Gini has a positive correlation with NTL Gini time series even in rural areas.

## Discussion

This study set out to evaluate whether night-time lights (NTL) data, particularly when operationalized as lights per capita at the pixel level, could serve as a meaningful proxy for measuring economic inequality at the municipal level. The primary research question was whether Gini coefficients calculated from NTL data correlate with those derived from administrative tax records on income. Both cross-sectional and longitudinal analyses were conducted to determine whether spatial and temporal variations in economic inequality could be captured by satellite imagery. By comparing two NTL sources—VIIRS and Harmonized datasets—with official income-based Gini coefficients from Spain, we aimed to assess both the reliability and the utility of remote sensing data for socioeconomic analysis around the world.

**Table 2. Bivariate linear regression results for Income-derived Gini and NTL Gini by population category.**

| NTL VIIRS | | | |
|---|---|---|---|
| **Pop category** | **Intercept** | **NTL estimate** | **NTL R²** |
| <= 2 | 29.29*** | 0.06 | 0.0000 |
| > 2 & <= 10 | 28.61*** | 1.41*** | 0.0075 |
| > 10 & <= 50 | 30.21*** | 1.22. | 0.0045 |
| > 50 & <= 100 | 33.47*** | −0.42 | 0.0004 |
| >100 & <= 250 | 28.17*** | 6.59. | 0.1234 |
| > 250 | 34.60*** | −0.31 | 0.0004 |
| NTL Harmonized | | | |
| **Pop category** | **Intercept** | **NTL estimate** | **NTL R²** |
| <= 2 | 29.24*** | 0.30 | 0.000 |
| > 2 & <= 10 | 29.53*** | −0.43 | 0.000 |
| > 10 & <= 50 | 31.89*** | −2.51** | 0.020 |
| > 50 & <= 100 | 32.61*** | 1.31 | 0.003 |
| >100 & <= 250 | 30.78*** | 4.82. | 0.082 |
| > 250 | 30.67*** | 7.40. | 0.212 |

(***) for highly significant results (p-value < 0.001), (**) for very significant results (p-value < 0.005), (*) for statistically significant results (p-value < 0.01), and a dot (.) for marginally significant results (p-value < 0.1).

The findings offer a nuanced yet promising perspective. Although cross-sectional models yielded modest correlations—with $R^2$ values typically below 0.01—the consistent and statistically significant associations observed in the panel regressions stand out. These longitudinal models demonstrate strong explanatory power across time and municipality sizes, revealing that changes in the spatial distribution of light are meaningfully linked to underlying economic transformations. This temporal consistency is perhaps the study's most significant contribution, underscoring the capacity of NTL-derived metrics to track inequality dynamics even in small or rural municipalities. Importantly, the alignment between NTL-based and income-based Gini coefficients—despite being derived from fundamentally different sources, each with known limitations—adds credibility to both. That two independent metrics converge over time suggests they are capturing shared, substantive dimensions of inequality.

The disparity in performance between the cross-sectional and longitudinal models can be interpreted as a reflection of how different factors influence light emissions. In cross-sectional analyses, lights-per-capita capture only a partial picture of socioeconomic variation, as light intensity is shaped not only by income levels but also by other structural elements such as infrastructure investment, energy usage norms, demographic composition, and urban design. These factors introduce variability that can weaken the relationship between lighting and income inequality across municipalities at a single point in time. In contrast, when examining changes within municipalities over time, many of these structural characteristics remain relatively constant. This temporal stability reduces confounding effects, allowing shifts in light distribution to more closely mirror real economic changes. The stronger performance of the panel models thus suggests that temporal variations in NTL data are more informative about evolving patterns of relative inequality. In this light, lights-per-capita should be interpreted less as precise indicators of inequality levels and more as dynamic signals of socioeconomic change, particularly useful where conventional economic data are sparse or lagged.

Our findings are consistent with previous studies that have linked NTL-based metrics to traditional measures of inequality [15–22]. Our study extends this work through a longitudinal, municipality-level analysis, demonstrating that NTL-derived Gini coefficients can effectively capture inequality dynamics over time and at fine spatial scales. However, our results diverge from those of Mirza et al. [16], who found significant cross-sectional correlations between light-based

**Table 3. Multilevel model results on NTL Gini and NTL from 2015 to 2020.**

| Model performance | | |
|---|---|---|
| **Metric** | **NTL VIIRS Model** | **NTL Harmonized Model** |
| AIC | 168362 | 169300.6 |
| BIC | 168421.9 | 169360.5 |
| Log-Likelihood | −84173.99 | −84643.3 |
| Random Effects | | |
| Std. Dev. (Intercept) | 3.44 | 3.10 |
| Std. Dev. (Slope: NTL) | 5.03 | 3.95 |
| Correlation (Intercept, Slope) | −0.57 | −0.32 |
| Residual Std. Dev. | 1.86 | 1.83 |
| Spatial Correlation Range | 1.18 | 1.09 |
| Fixed Effects | | |
| Intercept Value | 27.85 | 28.93 |
| Intercept Std. Error | 0.065 | 0.052 |
| Slope (NTL) Value | 4.33 | 2.41 |
| Slope (NTL) Std. Error | 0.135 | 0.130 |
| p-value (Intercept) | <0.001 | <0.001 |
| p-value (Slope: NTL) | <0.001 | <0.001 |
| Standardized Residuals | | |
| Min | −6.19 | −7.15 |
| Q1 | −0.52 | −0.54 |
| Median | −0.06 | −0.07 |
| Q3 | 0.46 | 0.49 |
| Max | 6.90 | 6.59 |
| Observations and Groups | | |
| Number of Observations | 38591 | 38585 |
| Number of Groups | 6610 | 6609 |
| Model R² | | |
| Marginal R² | 0.08 | 0.01 |
| Conditional R² | 0.75 | 0.73 |

and income-based inequality, but no significant relationship in longitudinal trends. Several methodological differences help explain this divergence and underscore the strength of our findings. First, their analysis relies mainly on DMSP-OLS data from before 2010, a satellite source with well-known limitations such as low spatial resolution, sensor saturation in brightly lit areas, and lack of onboard calibration. In contrast, we use VIIRS and Harmonized datasets, which provide higher radiometric sensitivity and temporal consistency, allowing for more reliable detection of changes over time. Second, Mirza et al. estimate inequality from the distribution of raw light intensity values, without accounting for population. Our use of lights per capita provides a more refined proxy, capturing relative differences in economic exposure and access. Third, their analysis operates at broad national and regional scales, where aggregation can obscure localized inequality dynamics. By focusing on municipalities, our study captures finer-grained spatial and temporal variation.

One unexpected outcome was the relatively better performance of the Harmonized NTL dataset in large urban municipalities with populations exceeding 250,000. This finding diverges from earlier studies [16] that generally favored VIIRS due to its higher spatial resolution. A likely explanation is sensor saturation in the VIIRS dataset: in highly illuminated urban centers, pixel values may reach a ceiling, limiting the differentiation between areas of varying economic intensity.

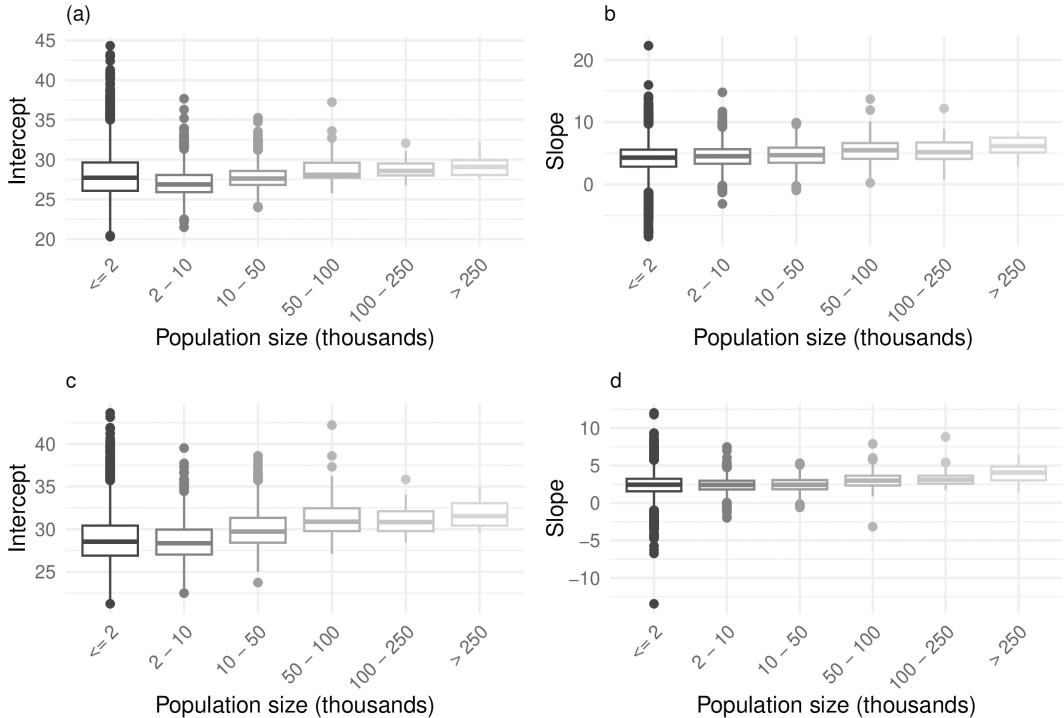

**Fig. 3. Intercept and slope values from multilevel random intercept and slope models using VIIRS and Harmonized NTL data, grouped by municipality population size.** The figure presents estimated intercepts and slopes for each population group, derived from multilevel models in which the income-based Gini coefficient is predicted by the NTL-derived Gini coefficient. Intercepts reflect baseline levels of inequality when NTL Gini is zero, while slopes indicate the strength and direction of the relationship between NTL-derived and income-derived Gini values. Results are shown separately for VIIRS and Harmonized datasets, with each estimate adjusted by adding municipality-specific random effects to the overall fixed effect—yielding population-weighted predictions for each group. Municipalities are categorized into six population brackets (e.g., ≤ 2,000; 2,001–10,000; … > 250,000) to illustrate how model performance varies across different levels of urbanization.

Harmonized data, with smoother transitions and longer time-series continuity, may be better suited to capturing subtle within-city inequalities.

Another unexpected outcome emerged in the geographic distribution of inequality. In southern Spain—particularly along the Guadalquivir axis in Andalusia, including cities such as Seville, Córdoba, and Jaén—NTL-derived Gini coefficients suggest markedly higher levels of spatial inequality than those reported in official income data. This discrepancy is especially pronounced in areas with historically high unemployment and lower levels of formal economic activity. One possible interpretation is that NTL data are capturing dimensions of informal or unregistered economic dynamics that are underreported or entirely absent in administrative income records. While visually compelling, these interpretations remain speculative and underscore the need for further research. Ground-level validation, ethnographic case studies, and integration with additional socioeconomic indicators could help disentangle the drivers behind such divergences and assess the validity of NTL-derived metrics as proxies for inequality in complex regional contexts.

The practical implications of our findings are substantial. NTL-derived inequality proxies can complement traditional economic statistics, especially in contexts where official data are delayed, incomplete, or unavailable. Remote sensing enables real-time, high-resolution monitoring of social conditions, offering valuable insights where conventional metrics fall short. Discrepancies between NTL- and income-based estimates should not be dismissed as errors; instead, they can highlight informal economies, uneven service provision, or disparities in infrastructure investment—factors often invisible in formal datasets. These insights can inform more inclusive and targeted policy interventions. The methodology is also

globally scalable: open-access NTL and population datasets allow replication in other countries, and historical data can be used to reconstruct long-term inequality trajectories. Finally, NTL data have clear applications for urban planning, helping to identify fine-grained spatial inequalities and guide equitable decisions in zoning, transport, and public service allocation.

### Identification of limitations and suggestions for prospective research

While promising, the methodological framework of this study entails several limitations. First, the use of WorldPop population estimates introduces a potential concern regarding collinearity, since nighttime lights are one of many covariates used in the population modelling. However, it is essential to emphasize that NTL data play a minor and supportive role in the WorldPop modelling process, which is primarily driven by census data, land cover, infrastructure, and accessibility variables [36]. In future work, alternative population datasets—such as those derived solely from census or household surveys—could be used to test the robustness of our results.

Second, sensor-related biases remain a challenge. VIIRS may saturate in highly lit areas, while Harmonized data may smooth over meaningful local variation. While unavoidable to some degree, this issue could be mitigated through contextual correction factors or by integrating ancillary geospatial datasets (e.g., land use or business registry data) to isolate non-income-related lighting.

Third, administrative income data used as the benchmark also suffer from known biases. The exclusion of informal income, pensions, and foreign remittances—especially relevant in southern Spain and among retired populations—means that official Gini coefficients may underreport actual inequality. The NTL proxy, by contrast, may capture some of this hidden variation, making the observed discrepancies informative rather than problematic. This limitation could be partly addressed in future work by complementing tax data with survey microdata or regional household accounts.

Lastly, while lights-per-capita is a useful operationalization, it is inherently limited. It does not capture within-building inequalities, differences in service access, or daytime economic activity. Moreover, how light is produced and distributed in different contexts is shaped by a wide array of social, cultural, and institutional factors that cannot be captured through satellite imagery alone. We suggest that future research incorporate qualitative and ethnographic approaches to better understand the local processes that give rise to observed light patterns. Such studies could shed light on how different sources of illumination—public infrastructure, private investment, or informal settlements—relate to, or diverge from, conventional indicators of economic well-being. By linking ground-level knowledge with satellite-based measurements, researchers can critically assess the meaning of light as a socioeconomic signal and avoid misinterpretations rooted in purely technical assumptions. Future research should consider integrating additional geospatial data such as housing quality, infrastructure access, or census microdata to build multi-dimensional proxies.

Prospective research should also explore the role of spatial segregation in modulating the effectiveness of NTL as an inequality proxy. As emphasized by Mirza et al. [20], the degree to which economic classes are spatially clustered affects how well light variation reflects income variation. Including segregation indices, such as entropy or dissimilarity metrics, could enhance explanatory power.

Moreover, enriching the model with variables related to urban morphology—such as building height, street connectivity, or land-use mix—could improve precision, especially in dense cities where light saturation and vertical development distort 2D spatial proxies. In addition, future work could benefit from engaging with studies that highlight how urbanization patterns, infrastructure expansion, and land-use policy shape spatial inequality [22,37–41]. Incorporating such dimensions may help to contextualize NTL-based metrics within broader processes of socioeconomic change, especially in rapidly transforming regions.

## Conclusion

This study examined whether nighttime lights (NTL) data, calculated as lights per capita at the pixel level, can serve as a robust proxy for measuring economic inequality at the local level in using Spain as our case study. Specifically, it assessed

whether NTL-derived Gini coefficients align with those based on official tax records and whether these remote-sensing indicators can reliably capture spatial and temporal variations in inequality.

The results demonstrate that NTL-derived Gini coefficients, particularly those generated using the VIIRS and Harmonized datasets, significantly correlate with income-derived Gini coefficients across time. Although cross-sectional correlations were modest, panel data models revealed strong temporal consistency, suggesting that NTL metrics are especially effective in tracking the dynamics of inequality. This relationship held across municipalities of all sizes, highlighting the method's applicability in both urban and rural contexts.

These findings validate the use of NTL as a valuable and scalable proxy for monitoring economic inequality, particularly in regions where conventional data may be delayed, incomplete, or unavailable. By offering high-resolution spatial and temporal insights, NTL-based indicators can complement traditional metrics, flag discrepancies that may reflect informal economies or infrastructural gaps, and inform more equitable policymaking. The study also opens avenues for historical inequality analysis through the integration of long-term NTL and gridded population datasets.

Future research should focus on refining NTL-based proxies by integrating additional geospatial variables—such as building density, access to services, or features of urban morphology—to enhance the accuracy of inequality estimates. Employing machine learning techniques or spatial econometric models could also improve the detection of non-linear patterns and spatial heterogeneity. Ground validation, particularly in rural or underrepresented areas, should be prioritized through targeted household surveys or the use of alternative satellite-derived socioeconomic indicators.

## Supporting information

**S1 Fig. Population Distribution of Municipalities in 2020, Categorized by Population Ranges (in thousands).** Source: Municipal register of inhabitants, National Institute of Statistics (INE). (PDF)

**S2 Fig. Evolution of the Gini Index across Different Data Types (2015–2020).** Each Gini estimate, except the National Gini (calculated from national-level income data), is derived from the average values across all Spanish municipalities. Source: Atlas of Household Income Distribution (ADRH)38 produced by the National Statistics Institute of Spain (INE). (PDF)

## Author contributions

**Conceptualization:** Xaquín S. Pérez-Sindín.

**Data curation:** Xaquín S. Pérez-Sindín.

**Formal analysis:** Xaquín S. Pérez-Sindín, Piotr Wójcik, Tzu-Hsin Karen Chen.

**Funding acquisition:** Xaquín S. Pérez-Sindín.

**Investigation:** Xaquín S. Pérez-Sindín, Piotr Wójcik.

**Methodology:** Xaquín S. Pérez-Sindín, Piotr Wójcik.

**Project administration:** Xaquín S. Pérez-Sindín.

**Resources:** Xaquín S. Pérez-Sindín, Piotr Wójcik, Alexander V. Prishchepov.

**Software:** Xaquín S. Pérez-Sindín.

**Supervision:** Xaquín S. Pérez-Sindín.

**Validation:** Xaquín S. Pérez-Sindín.

**Visualization:** Xaquín S. Pérez-Sindín, Piotr Wójcik, Tzu-Hsin Karen Chen.

**Writing – original draft:** Xaquín S. Pérez-Sindín.

**Writing – review & editing:** Xaquín S. Pérez-Sindín, Piotr Wójcik, Tzu-Hsin Karen Chen, Alexander V. Prishchepov.

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
