## [Decision Letter · Decision Letter 0]

4 Mar 2025

Dear Dr. Pérez-Sindín,

Thank you for submitting your manuscript to PLOS ONE. After careful consideration, we feel that it has merit but does not fully meet PLOS ONE’s publication criteria as it currently stands. Therefore, we invite you to submit a revised version of the manuscript that addresses the points raised during the review process.

We look forward to receiving your revised manuscript.

Kind regards,

Jun Yang

Academic Editor

PLOS ONE

Journal Requirements:

“The European Commission's Marie Skłodowska-Curie Individual Fellowship Action supported this work [Grant Agreement number: 101067663 — MAPSOCEXTRACT]. This work has also been supported by "Grant competition under Action I.3.9 "Human mobility and inequalities as seen through digital data sources", University of Warsaw.”

3. When completing the data availability statement of the submission form, you indicated that you will make your data available on acceptance. We strongly recommend all authors decide on a data sharing plan before acceptance, as the process can be lengthy and hold up publication timelines. Please note that, though access restrictions are acceptable now, your entire data will need to be made freely accessible if your manuscript is accepted for publication. This policy applies to all data except where public deposition would breach compliance with the protocol approved by your research ethics board. If you are unable to adhere to our open data policy, please kindly revise your statement to explain your reasoning and we will seek the editor's input on an exemption. Please be assured that, once you have provided your new statement, the assessment of your exemption will not hold up the peer review process

4. We note that Figure 1 and 2 in your submission contain map images which may be copyrighted. All PLOS content is published under the Creative Commons Attribution License (CC BY 4.0), which means that the manuscript, images, and Supporting Information files will be freely available online, and any third party is permitted to access, download, copy, distribute, and use these materials in any way, even commercially, with proper attribution. For these reasons, we cannot publish previously copyrighted maps or satellite images created using proprietary data, such as Google software (Google Maps, Street View, and Earth). For more information, see our copyright guidelines: http://journals.plos.org/plosone/s/licenses-and-copyright .

      a. You may seek permission from the original copyright holder of Figure 1 and 2 to publish the content specifically under the CC BY 4.0 license. 

Additional Editor Comments:

Reviewer 1

The authors explored the spatio-temporal inequality dynamics, the research is interesting. However, there have many problems need to be revised as follow.

1.The theme of this research is economic inequality. However, the economic is not shown in title. The title should be easily to read. such as spatio-temporal characteristic of economy inequality based on remote sensing data in Spain.

2.The English grammar should be noticed. In general, we use remote sensing data rather than remotely sensed data.

3.In line 145-147, how to shown age in your research?

4.In fact, the worldpop data is also inverted from NTL. How to solved the data homogeneity problem?

5.The content is insufficient. The conclusion and influence mechanism are not shown.

6.Although the article provides an explanation of the background of economic inequality, there is a lack of deeper discussion on the theoretical link between NTL data and economic inequality. For example, How can nighttime lighting reflect socioeconomic stratification? How does the wealth gap specifically affect light intensity distribution?

7.The study mentions "the removal of negative NTL data" but does not detail how these outliers are handled.

8.The article mentions that the NTL Gini coefficient in sparsely populated areas may better reflect informal economic activity, but does not further quantify this relationship. The heterogeneous should be mentioned.

9.The R² value of the cross-sectional regression model is low (e.g., the R² of VIIRS is mostly below 0.01), which indicates that NTL data have limited explanatory power for economic inequality.

10.The diagrams (Fig. 2 and Fig. 3) are briefly explained, and some important information is not fully explained.

11.Some relevant references are not shown in literature review. These research also used remote sensing data present economy or population.

Spatial evolution of population change in Northeast China during 1992–2018.Science of The Total Environment,2021,776,146023. doi:https://doi.org/10.1016/j.scitotenv.2021.146023

Spatiotemporal relationship characteristic of climate comfort of urban human settlement environment and population density in China. Frontiers in Ecology and Evolution,2022,10:953725. doi:10.3389/fevo.2022.953725

Ecosystem Service Value and Economy in the Pearl River Delta Urban Agglomeration of China.Land,2024, 13(10), 1670. doi: https://doi.org/10.3390/land13101670

Reviewer 2

The manuscript evaluates the potential of using night light (NTL) data to measure economic inequality at the municipal level in Spain. In addition, the authors combine Spain's comprehensive income registration data and diverse socioeconomic backgrounds to evaluate the effectiveness of NTL data in measuring inequality and explore its application prospects worldwide. The article is well structured and is recommended for publication after revision. My specific comments are as follows:

1. The abstract is the eye of the article. The article abstract needs to be further optimized and improved, and only needs to briefly introduce the background, methods, results and significance. In addition, research highlights need to be highlighted, such as the potential of using night light (NTL) data to evaluate economic inequality.

2. The introduction introduces research progress such as economic inequality, diverse socioeconomic backgrounds and night light (NTL) data, but it is not comprehensive enough. Discussions on socioeconomic development and imbalance should also consider aspects such as urbanization level, urban land use structure and development policies. It is recommended to refer to the following references to improve comprehensiveness. —Promoting or Inhibiting? Influence of Railway Container Transportation on Regional Economic Development Does land transfer promote the development of new-type urbanization? New evidence from urban agglomerations in the middle reaches of the Yangtze River� Spatio-temporal characteristics and influencing factors of land disputes in China: Do socio-economic factors matter?�Spatiotemporal Changes and Influencing Factors of the Coupled Production–Living–Ecological Functions in the Yellow River Basin, China Spatial heterogeneity of human settlements suitability from multidimensional perspectives in 31 provincial capital cities of China

3. In Table 1, the R² values of the VIIRS and Harmonized models are generally low (up to 0.03), which indicates that the NTL data has limited ability to explain the income Gini coefficient. It is necessary to supplement the analysis of this phenomenon in the discussion section, such as whether it is related to the spatial resolution of the NTL data or the accuracy of the population data.

4. When discussing the results of the panel data model, it is mentioned that the VIIRS model is slightly better than the Harmonized model in terms of goodness of fit. It is recommended to add a sentence about the advantages and disadvantages of the two models in practical applications, such as in which cases it is more appropriate to use VIIRS data and in which cases it is more appropriate to use Harmonized data.

5. When discussing the limitations of NTL data, it is mentioned that "NTL data may reveal higher levels of inequality in regions with a significant informal economy". Some specific methods on how to verify this hypothesis should be explained, such as whether it can be further verified through case studies or field surveys.

6. When discussing future research directions, a sentence can be added about how to integrate other geospatial data (such as urban density, infrastructure distribution, etc.) to improve the accuracy of the model.

7. The number of references in the article is insufficient. I suggest that the author add at least 15 relevant references to support the scientific and cutting-edge nature of the article.

Reviewers' comments:

Reviewer's Responses to Questions

**Comments to the Author**

1. Is the manuscript technically sound, and do the data support the conclusions?

Reviewer #1: Partly

Reviewer #2: Yes

2. Has the statistical analysis been performed appropriately and rigorously?

Reviewer #1: Yes

Reviewer #2: Yes

3. Have the authors made all data underlying the findings in their manuscript fully available?

Reviewer #1: Yes

Reviewer #2: Yes

4. Is the manuscript presented in an intelligible fashion and written in standard English?

Reviewer #1: No

Reviewer #2: Yes

Reviewer #1: The authors explored the spatio-temporal inequality dynamics, the research is interesting. However, there have many problems need to be revised as follow.

1.The theme of this research is economic inequality. However, the economic is not shown in title. The title should be easily to read. such as spatio-temporal characteristic of economy inequality based on remote sensing data in Spain.

2.The English grammar should be noticed. In general, we use remote sensing data rather than remotely sensed data.

3.In line 145-147, how to shown age in your research?

4.In fact, the worldpop data is also inverted from NTL. How to solved the data homogeneity problem?

5.The content is insufficient. The conclusion and influence mechanism are not shown.

6.Although the article provides an explanation of the background of economic inequality, there is a lack of deeper discussion on the theoretical link between NTL data and economic inequality. For example, How can nighttime lighting reflect socioeconomic stratification? How does the wealth gap specifically affect light intensity distribution?

7.The study mentions "the removal of negative NTL data" but does not detail how these outliers are handled.

8.The article mentions that the NTL Gini coefficient in sparsely populated areas may better reflect informal economic activity, but does not further quantify this relationship. The heterogeneous should be mentioned.

9.The R² value of the cross-sectional regression model is low (e.g., the R² of VIIRS is mostly below 0.01), which indicates that NTL data have limited explanatory power for economic inequality.

10.The diagrams (Fig. 2 and Fig. 3) are briefly explained, and some important information is not fully explained.

11.Some relevant references are not shown in literature review. These research also used remote sensing data present economy or population.

Spatial evolution of population change in Northeast China during 1992–2018.Science of The Total Environment,2021,776,146023. doi:https://doi.org/10.1016/j.scitotenv.2021.146023

Spatiotemporal relationship characteristic of climate comfort of urban human settlement environment and population density in China. Frontiers in Ecology and Evolution,2022,10:953725. doi:10.3389/fevo.2022.953725

Ecosystem Service Value and Economy in the Pearl River Delta Urban Agglomeration of China.Land,2024, 13(10), 1670. doi: https://doi.org/10.3390/land13101670

Reviewer #2: The manuscript evaluates the potential of using night light (NTL) data to measure economic inequality at the municipal level in Spain. In addition, the authors combine Spain's comprehensive income registration data and diverse socioeconomic backgrounds to evaluate the effectiveness of NTL data in measuring inequality and explore its application prospects worldwide. The article is well structured and is recommended for publication after revision. My specific comments are as follows:

1. The abstract is the eye of the article. The article abstract needs to be further optimized and improved, and only needs to briefly introduce the background, methods, results and significance. In addition, research highlights need to be highlighted, such as the potential of using night light (NTL) data to evaluate economic inequality.

2. The introduction introduces research progress such as economic inequality, diverse socioeconomic backgrounds and night light (NTL) data, but it is not comprehensive enough. Discussions on socioeconomic development and imbalance should also consider aspects such as urbanization level, urban land use structure and development policies. It is recommended to refer to the following references to improve comprehensiveness. —Promoting or Inhibiting? Influence of Railway Container Transportation on Regional Economic Development Does land transfer promote the development of new-type urbanization? New evidence from urban agglomerations in the middle reaches of the Yangtze River� Spatio-temporal characteristics and influencing factors of land disputes in China: Do socio-economic factors matter?�Spatiotemporal Changes and Influencing Factors of the Coupled Production–Living–Ecological Functions in the Yellow River Basin, China Spatial heterogeneity of human settlements suitability from multidimensional perspectives in 31 provincial capital cities of China

3. In Table 1, the R² values of the VIIRS and Harmonized models are generally low (up to 0.03), which indicates that the NTL data has limited ability to explain the income Gini coefficient. It is necessary to supplement the analysis of this phenomenon in the discussion section, such as whether it is related to the spatial resolution of the NTL data or the accuracy of the population data.

4. When discussing the results of the panel data model, it is mentioned that the VIIRS model is slightly better than the Harmonized model in terms of goodness of fit. It is recommended to add a sentence about the advantages and disadvantages of the two models in practical applications, such as in which cases it is more appropriate to use VIIRS data and in which cases it is more appropriate to use Harmonized data.

5. When discussing the limitations of NTL data, it is mentioned that "NTL data may reveal higher levels of inequality in regions with a significant informal economy". Some specific methods on how to verify this hypothesis should be explained, such as whether it can be further verified through case studies or field surveys.

6. When discussing future research directions, a sentence can be added about how to integrate other geospatial data (such as urban density, infrastructure distribution, etc.) to improve the accuracy of the model.

7. The number of references in the article is insufficient. I suggest that the author add at least 15 relevant references to support the scientific and cutting-edge nature of the article.

**Do you want your identity to be public for this peer review?** For information about this choice, including consent withdrawal, please see our Privacy Policy

Reviewer #1: No

Reviewer #2: No

---

## [Author Response · Author response to Decision Letter 1]

28 Jul 2025

Dear Editor and Reviewers,

We would like to express our sincere gratitude for your constructive and insightful comments on our manuscript. We have carefully addressed each suggestion and believe the revisions have significantly strengthened the clarity, depth, and overall quality of the paper. In the following pages, we provide detailed, point-by-point responses to all reviewer and editorial comments.

We have uploaded a clean version of the revised manuscript, a tracked-changes version, and a document with a point by point response to you comments.

Please don’t hesitate to let us know if further changes are needed. We thank you again for the opportunity to revise our work and for the constructive guidance received.

---

## [Decision Letter · Decision Letter 1]

6 Aug 2025

Dear Dr. Pérez-Sindín,

Thank you for submitting your manuscript to PLOS ONE. After careful consideration, we feel that it has merit but does not fully meet PLOS ONE’s publication criteria as it currently stands. Therefore, we invite you to submit a revised version of the manuscript that addresses the points raised during the review process.

We look forward to receiving your revised manuscript.

Kind regards,

Jun Yang

Academic Editor

PLOS ONE

Journal Requirements:

Additional Editor Comments:

Minor Revision

Reviewers' comments:

Reviewer's Responses to Questions

**Comments to the Author**

Reviewer #1: (No Response)

Reviewer #2: (No Response)

2. Is the manuscript technically sound, and do the data support the conclusions?

Reviewer #1: Yes

Reviewer #2: (No Response)

3. Has the statistical analysis been performed appropriately and rigorously?

Reviewer #1: Yes

Reviewer #2: (No Response)

4. Have the authors made all data underlying the findings in their manuscript fully available?

Reviewer #1: Yes

Reviewer #2: (No Response)

5. Is the manuscript presented in an intelligible fashion and written in standard English?

Reviewer #1: Yes

Reviewer #2: (No Response)

Reviewer #1: The authors have revised manuscript. However, there still have many problems need to be revised.

1.The title is a little exaggerate. The authors mainly explored the spatiotemporal characteristic of economy in Spain. The title could be revised to spatiotemporal characteristic economic in Spain:Based on the grid scale

2.The abstract is too long. It should be compressed to less than 350 words.

3.The keywords is mess, it should contain spatiotemporal characteristic, nighttime image, Spain.

4.In introduction, the time should be specific, in recent years is not specific.

5.The authors cited too many references. In general, one sentence cited 3 references at most. There have a sentences cited 18 references, it is too much.

6.The authors used nighttime image represent GDP or income? Most of research use nighttime image represent GDP. Is there have relevant references represent income ? In addition, the existing literature review is too less, some content should be moved to study area and data source(the fourth and fifth paragraph).

7.The structure is mess. In section and method, I only want to see three parts, study area, data source and methods. The authors should use less words to tell me your data source , and how to use it? The basic information is not necessary. I don't care the detailed information of nighttime light data.

8.The presentation is not clear. The research is grid scale or villages and towns scale� “A key factor in this process is the modified OECD scale, used to calculate income per consumption unit. This scale adjusts income based on household size and composition, assigning a weight of 1 to the first adult, 0.5 to additional members aged 14 or older, and 0.3 to those under the age of 14 [50]. In other terms, children are weighted less than adults when computing per-capita income.” It is difficult to understand and lack of invalidation.

9.The worldpop data is population density, how to transfer number of population, it should be illustrated.

10.In results, the sub-title is too long. The correlation between NTL and Income-derived Gini coefficients is too easy. The control variable is not shown. In addition, the authors only illustrated the data analysis, the reason of phenomenon is not mentioned.

11.The authors mentioned spatiotemporal characteristic and urban-rural disparities in your text. In fact, these content is not explained clearly. I am not sure your research emphasized the spatiotemporal characteristic or comparing two NTL sources—VIIRS and Harmonized datasets. The manuscript logic is poor. The relevant content is not shown in conclusion.

12.The authors listed too many limitations. All of them can not be solved?

Reviewer #2: The authors seriously addressed my concerns and I have no other comments. Therefore, I recommend publication of this promising version.

**Do you want your identity to be public for this peer review?** For information about this choice, including consent withdrawal, please see our Privacy Policy

Reviewer #1: No

Reviewer #2: No

---

## [Author Response · Author response to Decision Letter 2]

15 Aug 2025

Response to Reviewers – Manuscript ID: PONE-D-25-06682

Dear Editor and Reviewers,

We thank you for your constructive and insightful feedback. We have addressed each comment carefully, and believe the revisions have improved the clarity, depth, and overall quality of the paper. Below, we provide detailed point-by-point responses, along with both a clean and a tracked-changes version of the revised manuscript.

Response to Reviewer 1’s Comments

1. Title

We have updated the title to

“Can Nighttime Lights Serve as a Proxy for Economic Inequality at the Local Administrative Unit Scale? Evidence from Spain”

Which we believe better reflect the aim and scope of the article.

2. Abstract length

Thank you for pointing this out. The abstract has been reduced to fewer than 350 words while retaining the essential elements of background, methods, results, and implications.

3. Keywords

We appreciate your suggestion and have updated the keywords to include: spatiotemporal characteristic, nighttime lights, economic inequality, local administrative unit, remote sensing, Spain, inequality mapping.

4. Specific time reference

We have revised vague temporal expressions to be more specific. For example:

“Over the past two decades” instead of “In recent years”

5. Number of references

We have reduced the number of references per sentence to no more than three, following your recommendation.

6. NTL and income vs GDP

Our study evaluates the capacity of an NTL-based Gini coefficient to predict the conventional tax-statistics-based Gini for measuring economic inequality. The focus is therefore inequality, making income the appropriate benchmark rather than GDP. Literature in this specific subfield is limited, and the unresolved correlation between NTL- and income-based inequality measures is central to our paper. We have retained the referenced paragraphs in the Introduction as they justify Spain as a case study and outline our novel coefficient estimation, which the Methods section then develops in detail.

7. Methods structure

We have reorganized the Methods section into three subsections: Study Area, Data Sources, and Methods. We retained concise descriptions of each dataset because their limitations are important for assessing the reliability of our results. However, we have reduced unnecessary detail.

8 Scale

We have emphasized in the title, abstract, and text that our analysis is at the local administrative unit (municipality) scale, using indicators of inequality calculated based on all pixels within each municipality.

9. The worldpop data is population density, how to transfer number of populations, it should be illustrated.

The worldpop data is actually population count. We have now indicated it clearly in the text.

10. Control variables and phenomenon explanation

The aim of this study is to assess whether nighttime-light-based Gini coefficients correlate with those derived from tax statistics. While existing literature has consistently shown a strong relationship between NTL and economic activity, only a few studies—those to which we seek to contribute—have examined whether NTL can also predict inequality. Using Spain as a case study and working at an unprecedented level of detail (municipality rather than regional scale), our objective is to produce a validated indicator of inequality for each municipality.

11. Focus of the study

We agree that urban–rural disparities are not a central focus and have removed them from the keywords. The manuscript now emphasizes the main contribution: validating NTL as a proxy for inequality and comparing VIIRS and Harmonized datasets.

12. Limitations

We appreciate the reviewer’s observation regarding the number of limitations listed. In the revised manuscript, we have streamlined the “Identification of Limitations” section to focus on the most relevant constraints of our approach and to distinguish between those that could be addressed in future work and those inherent to the methodology.

Specifically, we removed the limitation related to potential noise from commercial activity and other non-residential lighting sources, as this issue is already addressed in the Methods section through the exclusion of all pixels without population counts. This preprocessing step effectively mitigates the bias from uninhabited but brightly lit areas.

We also removed the point on accounting for spatial autocorrelation. While such modelling refinements can be valuable in some contexts, their contribution to the main objective of this study—validating nighttime lights as a proxy for inequality—is uncertain. For prudence and clarity, we decided not to introduce methodological extensions that may not directly serve our primary research goal.

The final limitations section now emphasizes four key issues: (1) the role of WorldPop in population estimation, (2) sensor-related biases, (3) known biases in administrative income data, and (4) the conceptual limitations of lights-per-capita as a socioeconomic indicator. For each, we note whether and how it could be addressed in future research. We believe this revised, more focused discussion better addresses the reviewer’s concern while preserving transparency.

---

## [Decision Letter · Decision Letter 2]

6 Sep 2025

PONE-D-25-06682R2Can Nighttime Lights Serve as a Proxy for Economic Inequality at the Local Administrative Unit Scale? Evidence from SpainPLOS ONE?

Dear Dr. Pérez-Sindín,

Thank you for submitting your manuscript to PLOS ONE. After careful consideration, we feel that it has merit but does not fully meet PLOS ONE’s publication criteria as it currently stands. Therefore, we invite you to submit a revised version of the manuscript that addresses the points raised during the review process.

We look forward to receiving your revised manuscript.

Kind regards,

Jun Yang

Academic Editor

PLOS ONE

Journal Requirements:

Additional Editor Comments:

Reviewer #1: The revised manuscript is not make me satisfied. The reasons are as allow.

1.The abstract is still too long. In fact, many words are not important enough.The author lacks of the ability to summarize. Such as Spain is among the most unequal EU countries by the Gini index and has published municipal income distribution records since 2015. Its geographic and demographic diversity— from dense metropolitan areas to sparsely populated rural municipalities—provides a robust setting for testing NTL-based measures with broader applicability, including in countries with limited urbanization.

2.Over the past two decades is not specific yet. The specific years should be mentioned.

3.I have mentioned one sentence should not cited 12 references. [12-24][25-32]. This problems still not revised.

4.The authors explored the across-sectional bivariate lineal regression analysis for the year 2015, 2016, 2017, 2018, 2019 and 2020 and one panel-data regression model for the study of temporal inequality dynamics between 2015 and 2020. What do similarities and differences indicate, and what are their implications?

5.The authors introduced the data detailed. The important information is not shown, such as website.

6.In fact, worldpop is also origin from night lighttime data. They have inherently consistent.

Reviewer's Responses to Questions

**Comments to the Author**

Reviewer #1: (No Response)

Reviewer #2: (No Response)

2. Is the manuscript technically sound, and do the data support the conclusions?

Reviewer #1: Yes

Reviewer #2: (No Response)

3. Has the statistical analysis been performed appropriately and rigorously?

Reviewer #1: Yes

Reviewer #2: (No Response)

4. Have the authors made all data underlying the findings in their manuscript fully available?

Reviewer #1: Yes

Reviewer #2: (No Response)

5. Is the manuscript presented in an intelligible fashion and written in standard English?

Reviewer #1: Yes

Reviewer #2: (No Response)

Reviewer #1: The revised manuscript is not make me satisfied. The reasons are as allow.

1.The abstract is still too long. In fact, many words are not important enough.The author lacks of the ability to summarize. Such as Spain is among the most unequal EU countries by the Gini index and has published municipal income distribution records since 2015. Its geographic and demographic diversity— from dense metropolitan areas to sparsely populated rural municipalities—provides a robust setting for testing NTL-based measures with broader applicability, including in countries with limited urbanization.

2.Over the past two decades is not specific yet. The specific years should be mentioned.

3.I have mentioned one sentence should not cited 12 references. [12-24][25-32]. This problems still not revised.

4.The authors explored the across-sectional bivariate lineal regression analysis for the year 2015, 2016, 2017, 2018, 2019 and 2020 and one panel-data regression model for the study of temporal inequality dynamics between 2015 and 2020. What do similarities and differences indicate, and what are their implications?

5.The authors introduced the data detailed. The important information is not shown, such as website.

6.In fact, worldpop is also origin from night lighttime data. They have inherently consistent.

Reviewer #2: The author carefully revised this article and addressed my concerns. I have no further comments. If other reviewers have no further concerns, I suggest publishing this version.

**Do you want your identity to be public for this peer review?** For information about this choice, including consent withdrawal, please see our Privacy Policy

Reviewer #1: No

Reviewer #2: No

---

## [Author Response · Author response to Decision Letter 3]

6 Sep 2025

Response to Reviewer 1’s Comments

1. Abstract length

We thank the reviewer for this observation. The abstract has now been significantly shortened to under 300 words, following the PLOS ONE guidelines. Redundant phrasing and descriptive details (such as Spain’s relative inequality and geographic diversity) were removed or streamlined to improve conciseness and focus.

2. Time specification (“Over the past two decades”)

We agree with the reviewer that the original phrasing was too vague. This has been revised to: “Following the early 2000s financial liberalization and the 2008 global financial crisis…”, which provides a more precise temporal anchor.

3. Excessive citations in one sentence

We acknowledge the reviewer’s point and have reduced the number of references per sentence to 4–5, except in one case ([25–32]) where all cited sources are essential.

4. Cross-sectional vs. panel regression implications

We appreciate this request for clarification. The implications of similarities and differences between the cross-sectional and panel models are now explicitly highlighted in both the abstract and the conclusions. In particular, we emphasize that panel models capture temporal dynamics more effectively, while cross-sectional regressions highlight spatial patterns at given points in time.

5. Data sources and missing website links

We thank the reviewer for pointing this out. All datasets now include complete references, with URLs/websites added where they were previously missing.

6. WorldPop dependence on nighttime lights

We acknowledge this concern and reiterate that it was addressed in the revised discussion. We clarified that while NTL data are included among the covariates in the WorldPop modelling process, their role is minor and supportive compared to census, land cover, infrastructure, and accessibility data. We also added that alternative population datasets (solely derived from census or surveys) could be used in future research to test robustness.

---

## [Decision Letter · Decision Letter 3]

28 Oct 2025

Can Nighttime Lights Serve as a Proxy for Economic Inequality at the Local Administrative Unit Scale? Evidence from Spain

PONE-D-25-06682R3

Dear Dr. Pérez-Sindín,

We’re pleased to inform you that your manuscript has been judged scientifically suitable for publication and will be formally accepted for publication once it meets all outstanding technical requirements.

**Comments from the PLOS Editorial Office** : We note that one or more reviewers has recommended that you cite specific previously published works in an earlier round of revision. As always, we recommend that you please review and evaluate the requested works to determine whether they are relevant and should be cited. It is not a requirement to cite these works and you may remove them before the manuscript proceeds to publication. We appreciate your attention to this request.

Kind regards,

Guy J-P. Schumann

Section Editor

PLOS ONE

Additional Editor Comments (optional):

Reviewers' comments:

Reviewer's Responses to Questions

**Comments to the Author**

Reviewer #2: (No Response)

2. Is the manuscript technically sound, and do the data support the conclusions?

Reviewer #2: (No Response)

3. Has the statistical analysis been performed appropriately and rigorously?

Reviewer #2: (No Response)

4. Have the authors made all data underlying the findings in their manuscript fully available?

Reviewer #2: (No Response)

5. Is the manuscript presented in an intelligible fashion and written in standard English?

Reviewer #2: (No Response)

Reviewer #2: The author has carefully revised the manuscript according to my suggestions and addressed my concerns. I recommend publication of this version.

**Do you want your identity to be public for this peer review?** For information about this choice, including consent withdrawal, please see our Privacy Policy

Reviewer #2: No

---

## [Editor Report · Acceptance letter]

PONE-D-25-06682R3

PLOS ONE

Dear Dr. Pérez-Sindín,

I'm pleased to inform you that your manuscript has been deemed suitable for publication in PLOS ONE. Congratulations! Your manuscript is now being handed over to our production team.

Kind regards,

on behalf of

Dr. Guy J-P. Schumann

Section Editor

PLOS ONE